# How Effective Is Phage Therapy for Prosthetic Joint Infections? A Preliminary Systematic Review and Proportional Meta-Analysis of Early Outcomes

**DOI:** 10.3390/medicina60050790

**Published:** 2024-05-09

**Authors:** Jason Young, Nicita Mehta, Sang Won Lee, Edward Kenneth Rodriguez

**Affiliations:** 1Harvard Combined Orthopedic Residency Program, Boston, MA 02114, USA; 2Harvard Medical School, Harvard University, Boston, MA 02115, USA; 3Carl J Shapiro Department of Orthopaedics, Beth Israel Deaconess Medical Center, Boston, MA 02215, USA

**Keywords:** bacteriophage, phage therapy, prosthetic joint infection, experimental therapeutic, early outcomes, meta-analysis

## Abstract

*Background and Objectives:* Despite the promise of phage therapy (PT), its efficacy in prosthetic joint infection (PJI) management is unknown. Much of the current literature is largely limited to case reports and series. *Materials and Methods:* In order to help inform power calculations for future clinical trials and comparative analyses, we performed a systematic review and proportional meta-analysis of early PT outcomes to provide a preliminary assessment of early phage therapy treatment outcomes for cases of PJI. *Results:* In a search of available literature across MEDLINE (Ovid, Wolters Kluwer, Alphen aan den Rijn, The Netherlands), Embase (Elsevier, Amsterdam, The Netherlands), the Web of Science Core Collection (Clarivate, London, UK), and Cochrane Central (Wiley, Hoboken, NJ, USA) up to 23 September 2023, we identified 37 patients with PJIs receiving adjunctive PT. Patients most frequently reported *Staphylococcal* species infection (95%) and intraarticular phage delivery (73%). Phage cocktail (65%) and antibiotic co-administration (97%) were common. A random-effects proportional meta-analysis suggested infection remission in 78% of patients (95% CI: 39%, 95%) (*I*^2^ = 55%, *p* = 0.08) and 83% with a minimum 12-month follow-up (95% CI: 53%, 95%) (*I*^2^ = 26%, *p* = 0.26). *Conclusions:* Our study provides a preliminary estimate of PT’s efficacy in PJIs and informs future comparative studies.

## 1. Introduction

Prosthetic joint infections (PJIs) are a significant cause of morbidity and mortality for patients after total joint replacement [1] The incidence of PJIs varies, with reported rates above 2% of all arthroplasties performed in some studies [1,2]. The principal challenge in eradicating PJIs, particularly chronic infections, is the formation of bacterial biofilms on the implanted prosthesis, which serve as microenvironments that promote infection persistence and recurrence as well as resistance to antimicrobial therapy [2,3].

Treating PJIs is associated with significant costs, morbidity, and dysfunction [2]. Direct in-patient medical costs alone to treat a single case are estimated at over $50,000 in the United States, twice as much for aseptic arthroplasty revision [4]. Outcomes also remain poor, with reported successful eradication rates under 80% for a two-stage revision [1]. Given the projected increase in the number of arthroplasties by 2030 [5], the development of new treatment strategies to more effectively and safely eradicate these infections represents an opportunity to decrease costs, decrease surgical morbidity, and improve overall patient outcomes.

Phage therapy, or the application of bacteriophages with the goal of infection elimination, has gained interest as a potential adjunctive therapy for the treatment of PJIs [6,7]. Well tolerated in vivo [6,7,8,9,10], many phages possess antibiofilm properties [10,11,12], and many have been noted to work synergistically with antibiotics to eliminate biofilm infections [12,13]. However, there is ongoing uncertainty regarding the efficacy of phage therapy when applied in PJI cases; current reports have largely been confined to case reports [14,15,16] or small case series [17,18,19], and to the authors’ knowledge, a summative analysis of in-human infection eradication rates after phage therapy application for PJI has not been performed. Despite this uncertainty, there is a need for larger, comparative studies, and there is now a phase 2 clinical trial underway to assess the therapeutic potential of phages for PJI management [20]. As such, a preliminary summary of the best available evidence is needed to inform the recruitment and design of this trial as well as serve as an effect estimate to inform power calculations of future comparative studies.

Consequently, we conducted a systematic review and proportional meta-analysis to provide a preliminary assessment of early phage therapy treatment outcomes for cases of PJI. Our principal aim is not to provide a comprehensive assessment of PT treatment effect. Rather, we hope this analysis serves as a review of the available early literature documenting application of phage therapy for PJI and enable us to obtain a first aggregate estimate of phage therapy efficacy to inform further clinical research. Our hope is that this assessment may inform statistical power calculations, help determine recruitment thresholds for trials, and ultimately contextualize findings of ongoing and future comparative studies.

## 2. Materials and Methods

Our systematic review was conducted in accordance with previously published guidelines [21] and reported in accordance with the PRISMA Reporting Guidelines [22]. This study was also registered in PROSPERO (ID: CRD42023467827) [23]. We conducted a literature review, screening, and assessment of included studies reporting on treatment success or failure of phage application to treat PJIs. PRISMA checklists for the systematic review and associated abstract are provided in Appendix A.

Our primary outcome was the proportion of patients achieving early infection remission as reported by a study’s authors, for which we defined early infection remission broadly as resolution of clinical signs and symptoms of infection, with or without culture confirmation. Given the preliminary nature of available reports, true PJI cure (which has previously been defined by the clinical or microbiologically tested absent infection without need for ongoing antibiotics at 24 months) [24] was not assessed given expected inadequate follow-up in the available preliminary reports. However, in addition to combined early infection remission rates calculated from author-reported follow-up endpoints, we assessed early infection remission at 12 months in a subgroup analysis. As the standard of care often involves administration of suppressive antibiotic therapy for patients after receiving debridement antibiotics and implant retention (DAIR) [25,26], the need for continued suppressive antibiotics at time of last follow-up did not preclude categorization of a successful early infection remission.

All phages, phage combinations, and phage and antibiotic combination therapies were included for all prosthetic joint infections involving knee and hip arthroplasties. Included treatments comprised preoperative, intraoperative, or postoperative phage therapy for PJI. As this is an emerging technology, our comparator group for studies that reported a comparator was the standard of care at the time a study was conducted. This comparator comprised either two-stage revision with irrigation and debridement, single-stage revision with irrigation and debridement, or DAIR. All included studies reporting on treatment success were aggregated for our meta-analysis.

### 2.1. Review Question

For patients with prosthetic joint infections involving the hip and knee being treated with phage therapy as an adjunctive or salvage therapy, what was the proportion of patients for which early infection eradication achieved?

### 2.2. Eligibility Criteria

Inclusion Criteria: We included all peer-reviewed studies in English reporting on a series of patients (*n* ≥ 2) undergoing treatment as part of a defined protocol with culture confirmed prosthetic joint infection (hip or knee arthroplasty) who receive phage therapy as part of their infection eradication treatment.

Exclusion Criteria: We excluded individual case reports, phage characterization studies, conference abstracts, erratum, opinion pieces, and retracted studies. We excluded individual case reports in order to focus solely on published literature of patients who had undergone treatment as part of a defined center’s experience or as part of a pre-defined treatment protocol in order to limit the number of one-off reports which may be subject to greater publication bias.

### 2.3. Search Strategy

Articles were extracted from MEDLINE (Ovid, Wolters Kluwer, Alphen aan den Rijn, The Netherlands), Embase (Elsevier, Amsterdam, The Netherlands), the Web of Science Core Collection (Clarivate, London, UK), and Cochrane Central (Wiley, Hoboken, NJ, USA). We began with a preliminary search in PubMed to identify keywords and MeSH terms before developing a final set of queries (Appendix A). A search was conducted on 23 September 2023. Article titles and abstracts were imported into Covidence (Melbourne, Australia) for data management. All titles and abstracts were screened independently by two authors (JY and SL) based on the inclusion and exclusion criteria. Conflicts were resolved through consensus. If a consensus could not be reached, a third author was used to determine inclusion or exclusion for any remaining conflicts (NM). Citations were managed through Endnote (Clarivate).

### 2.4. Data Extraction

Articles selected for inclusion were evaluated in full-text using a data extraction tool by 2 authors (JY and NM) (Appendix A). Data collected included study country, study design number of participants, demographic information, infecting organism and infected total joint, whether patients had failed prior conventional treatment, phage delivery mechanism, therapy duration, and any concomitant antibiotics or surgery, as well as length of follow-up and documented treatment outcome and complications. All results for the outcome were sought and evaluated. Disagreements were resolved through discussion by both authors. Corresponding authors were contacted for any missing data or for clarification of questions.

### 2.5. Quality Assessment

Two authors (JY and NM) independently performed a bias assessment using the NHL Study Quality Assessment Tools [27]. This tool was employed due to its accommodation of case series [27] which constituted a majority of included studies. Any disagreements were discussed between authors on article review until a consensus was reached. Quality ratings were based on an overall assessment of the assessed work and also determined based on consensus.

### 2.6. Data Synthesis

All included articles for which authors reported on the proportion of patients achieving infection remission, as defined through achieving at least one of our endpoints in our composite definition of our primary outcome, were deemed eligible for synthesis. We set a minimum threshold of 3 studies and a maximum *I*^2^ heterogeneity estimate of 80% to proceed with a meta-analysis calculation. We specified *a priori* a minimum of 10 studies to perform an Egger test and funnel plots to assess for publication bias. No additional data conversions were necessary for data synthesis or presentation. Data collected from all included studies were separately summarized and tabulated.

### 2.7. Statistics

We performed a random effects proportional meta-analysis examining the proportion of author-reported treatment success of recalcitrant PJI after phage therapy using an inverse-variance weighted average. We chose a random effects model given the variety in anatomic location of infection and causative organism, as well as variety in phage regimen delivery methods, dosing schedules, and treatment durations. Heterogeneity was estimated using *I*^2^ with an alpha set at 0.10, and the confidence interval for the summary effect was estimated using the Wald-type confidence interval method. A funnel plot and Egger’s test was to be constructed to assess for publication bias if at least 10 studies were included in our meta-analysis. Sensitivity analyses, descriptive statistics, and proportional meta-analyses were conducted in R (RStudio, version 2022.12.0+353, Posit Software, PBC, Boston, MA, USA).

### 2.8. Certainty Assessment

The preliminary nature of the research findings and limited number of comparative studies available led us to defer a certainty of evidence assessment based on GRADE guidelines [28].

## 3. Results

A summary of our search is depicted in Figure 1. In summary, 745 unique articles were screened in this study, with 736 articles removed on screening and 9 articles meeting criteria to be evaluated on full text review. Five studies were excluded, yielding four studies which were included in the final analysis [19,25,26,29]. Of the five excluded studies, one presented duplicate data from another study included in the analysis, one did not describe original data, one case report was excluded according to exclusion criteria, one study described a series of patients for which only a single PJI case was described, and one study did not describe phage applications for PJIs.

### 3.1. Study Characteristics

Studies included in this meta-analysis are summarized in Table 1. Briefly, case series comprised three of the four included studies [19,25,26], while a single case–control study was also included [29]. In total, 37 patients received phage therapy. Phages were administered for the management of either prosthetic hip infection (*n* = 24), prosthetic knee infection (*n* = 10), both prosthetic hip and knee infections (*n* = 2), or for an unspecified total joint arthroplasty (*n* = 1). In terms of associated surgical procedure, debridement, antibiotics, and implant retention (DAIR) was attempted in nine patients, a single-stage revision attempted in 23 patients, while a single-stage or two-stage revision attempted in four patients (unspecified surgical procedure for 2).

The implicated bacterial organism confirmed via culture for all patients is presented in Figure 2. Staphylococcal species predominated in 35 of 37 patient infections (95%), with 6 (16%) cases of methicillin-resistant *Staphylococcus aureus* (MRSA), 12 (32%) cases of methicillin-sensitive *Staphylococcus aureus* (MSSA), 15 (41%) cases of *Staphylococcus epidermidis* infections, and 2 (5%) cases of *Staphylococcus lugdunensis* infection. Two (5%) cases of *Enterococcus faecalis* infection were also reported.

### 3.2. Quality Assessment

The results of our quality assessment evaluation of included studies is presented in Appendix A. Three (75%) included studies were given a “Fair” rating and one (25%) a “Poor” rating.

### 3.3. Study Findings

In terms of delivered phage therapy, intraarticular phage was delivered in 27 (73%) patients, combined intraarticular and intravenous delivery employed for 8 (22%) patients, and intravenous delivery used in 2 (5%) patients. Monophage delivery was used in 10 (27%) patients, while 24 (65%) patients received a phage cocktail (unspecified in 3 patients). Therapy duration ranged from a single dose immediately at the end of surgical debridement to 45 days. Co-administration of antibiotics was performed in 36 (97%) patients. Long-term suppressive antibiotics were administered to 8 (22%) patients.

Follow-up ranged from 3 months to 30 months. Adverse events related to phage therapy included transient fevers in two patients and asymptomatic transaminitis in five patients. No other adverse events were reported.

### 3.4. Results of Synthesis

The results of our meta-analysis of the proportion of phage therapy patients with author-reported infection remission are presented in Figure 3. Based on our random-effects model, the aggregate proportion of patients achieving clinical remission of their infection was 0.78 (95% confidence interval, CI: 0.39, 0.95) (*I*^2^ = 55%, *p* = 0.08) with evidence of moderate heterogeneity.

Given the short follow-up reported for some patients in the included studies, a subgroup analysis of only patients with at least 12 months follow-up post-phage delivery was performed, with results reported in Figure 4. Based on a random-effects model, the aggregate proportion of patients with at least 12 months follow-up achieving clinical remission of their infection was 0.83 (95% CI: 0.53, 0.95) (*I*^2^ = 26%, *p* = 0.26) without evidence of statistically significant heterogeneity.

Given the limited number of included studies, an evaluation of reporting bias, sensitivity analyses, and further exploration of statistical heterogeneity were not performed.

## 4. Discussion

The rise of antibiotic-resistant bacterial organisms has proven to be a clinical challenge and accelerated the search for alternative forms of antibiosis, and among these alternatives, phage therapy has demonstrated unique potential in the elimination of biofilms and PJIs [30,31]. As interest in phage therapy has grown, application of this technology in the treatment of PJIs has become an area of active research. A dedicated preliminary summary of currently published work detailing outcomes of phage therapy can serve to inform power calculations for comparative studies and contextualize future comparative work examining phage therapy treatment efficacy for PJIs. Our proportional random-effects meta-analysis suggests that the approximate proportion of patients achieving infection remission after surgical debridement, antibiotics, and adjunctive phage therapy to be 0.78, and 0.83 for patients with at least 12 months follow-up and represents one of the first estimates of infection eradication after application of phage therapy for PJIs.

To the authors’ knowledge, this proportional meta-analysis represents the first dedicated estimate of early infection remission from PJI after phage therapy. We report a pooled early infection remission proportion of 0.78 (78%) based on all available patients and 0.83 (83%) for those with at least 12 months follow-up. Our findings align with other estimates of infection eradication after phage therapy. In their systematic review of phage therapy, safety, and efficacy of phage therapy for bone and joint infections, Clarke et al. described results from 277 patients, of which 229 (83%) were patients treated for osteomyelitis, and report a crude estimate of 93% clinical infection resolution in the 17 articles included [6]. Separately, Genevière et al. reported in their systematic review of phage therapy for bone and joint infections 52 phage treatments in 51 patients, of which 44 (85%) applications were topical, reporting an overall success rate of 71%, though these authors do note a success rate in PJI of 57% [32]. Finally, in their systematic review of superficial bacterial infection treatments of skin, burns, and chronic wound infections treated with phage therapy, Steele et al. reported a pooled infection improvement or resolution rate of 77.5% of burns, 86.1% of chronic wounds, and 94.1% of skin infections [33]. The relatively high rate of infection eradication after use of phage therapy across these varied indications supports its therapeutic potential as a new class of antibiosis in the management of intransigent infections.

Our results highlight the clinical potential of phage therapy for PJI management. Though preliminary analyses, our estimate would suggest that phage therapy, when used adjunctively with surgery and antibiotics, may exhibit infection eradication rates for PJI at high enough levels as to be comparable to the current gold-standard two-stage revision, for which various reports in the literature have cited infection eradication rates between 75% and 90% [1,34,35,36,37]. This is particularly notable as at least 12/37 (32%) of patients included in this analysis were documented to have failed previous surgical and antibiotic therapies. Unfortunately, the lack of comparative studies limits further relative comparisons between phage therapy and two-stage revisions, or how infection remission rates might differ between surgical procedures (DAIR versus one-stage revision versus two-stage revision) when phages are also applied. While in vivo evidence in humans remains limited, a clinical trial has recently been registered in which phage therapy will be used as an adjunctive treatment for infection eradication [20], and they may provide much needed comparative evidence examining the treatment effect of phage therapy. We hope that the proportional estimate of infection eradication provided in this report can serve as a preliminary estimate for adjunctive phage therapy and serve as a component for power calculations and sample size estimates in these trials and other future comparative studies examining phage therapy for PJI management.

Interestingly, our meta-analysis would suggest only moderate heterogeneity (*I*^2^ of 0.55 and *p* = 0.08) for our studied outcome, and only for all reported patients (no significant heterogeneity found among patients with at least 12 months follow-up) despite a variety of delivered phages, associated surgical procedures, delivery strategies, and therapy durations described. Explanatory mechanisms remain elusive, but may relate to a permissiveness phage to delivery strategy [38] and dosing [39] with respect to generation of an observable clinical effect, possibly through local expansion of phage numbers at sites of infection [39], as long as delivery is successful. This theory would be supported by the wide range of administration routes and treatment durations previously described [32]. Understanding whether this permissiveness truly exists in practice will require higher-level evidence and larger patient cohorts to develop a more granular understanding of relative bacterial eradication rates achieved through different delivery and dosing protocols.

Certainly, our study is not without limitations. First, the low number of patients and limited number of included studies limits interpretation and clinical utility beyond serving as a preliminary estimate. Indeed, the preponderance of case series in our meta-analysis and lack of high-level evidence places our infection eradication estimate at risk for reporting bias, and the low number of included studies limits our ability to statistically assess for this bias. Our estimate may consequently be an overestimation of phage therapy’s infection eradication capacity. However, the purpose of this meta-analysis is to provide a preliminary estimate based on the best available evidence, and our estimate can thus be interpreted as a potential upper limit of treatment efficacy. Future studies involving well-controlled comparator groups, carefully selected indications and well-documented phage treatment protocols, clearly defined outcomes, as well as larger patient cohorts will be needed to obtain a more accurate summative estimate of phage therapy’s treatment effect. Second, due to the limited follow-up of available studies, as well as the use of suppressive antibiotics for patients receiving DAIR [25,26], we are unable to definitively document true infection cure rates after phage therapy administration. We consequently report both author-reported early infection remission rates as well as remission rates at 12 months follow-up as a means of estimating early treatment efficacy of phage therapy. As additional studies are published and longer-term follow-up established, a more accurate assessment of phage therapy and cure rates can be calculated. Third, our outcome of infection remission is dependent on follow-up time as well as the clinical and laboratory assessments performed by the various authors of our included studies and thus is dependent on the accuracy, definitions, and timescales of our included studies. Future studies can consider adopting a standardized definition or at minimum, similar endpoints to measure infection eradication to facilitate comparability between studies. Fourth, the studies included in our analysis report on a variety of treatment protocols which, while not generating a statistically significant degree of heterogeneity with respect to our proportional infection eradication estimate, does limit study comparability. This variability may be attributable to a lack of established treatment protocols described for phage therapy applications in humans [40]. Future work is needed to develop and assess treatment regimens, taking into account implicated bacterial species, infection chronicity, prior treatment history, bacterial sensitivity profile to antibiotics, phages used, patient comorbidities, treatment duration, and phage delivery method. Fifth, Staphylococcal species accounted for the majority (95%) of PJIs included in this meta-analysis, and consequently, caution is required when considering application of our estimate to Gram-negative and anaerobic PJI cases. Sixth, the number of available studies included in our meta-analysis is small, limiting our ability to perform subgroup investigations and bias assessments. To date, reports of phage therapy application for PJI treatment remain limited, as its use in clinical practice is confined to cases of compassionate use in the Western world [41]. As such, there remains a lack of large-scale studies whose publication will be necessary for further advancement of clinical understanding of phage therapy’s effects in practice. Finally, we decided to exclude all individual case reports from our analysis. While this may further lower the number of included patients in this meta-analysis, we believed exclusion of single reports helps limit publication bias, allowing us to focus on patients who have undergone treatment as a part of a defined treatment protocol at an institution with PT experience. Despite these limitations, the findings presented in this study represent a preliminary summary of phage therapy efficacy for PJI management and may serve to inform power calculations for larger studies and contextualize results of future clinical applications.

## 5. Conclusions

As interest in phage therapy has grown, application of this technology in the treatment of PJIs has become an area of active research. A dedicated preliminary summary of currently published work detailing outcomes of phage therapy can serve to inform power calculations for comparative studies and contextualize future comparative work examining phage therapy treatment efficacy for PJIs. Our proportional random-effects meta-analysis suggests adjunctive use of phages can help elicit an approximate infection remission rate of 80% in the studies included in this analysis, in line with prior estimates of infection eradication rates for musculoskeletal infections and comparable to documented infection eradication rates for two-stage revisions. The lack of statistically significant heterogeneity despite variation in administered phage regimens may suggest a permissiveness of effect with respect to dosing and delivery strategy, which will require future study. Ultimately however, this work only represents a preliminary estimate of early PT outcomes for PJI, limited by the small number of patients and the paucity of published studies. As PT gains traction as a treatment, additional higher-level studies are needed to further explore PT and better understand its role for PJI management as a whole.

## Figures and Tables

**Figure 1 medicina-60-00790-f001:**
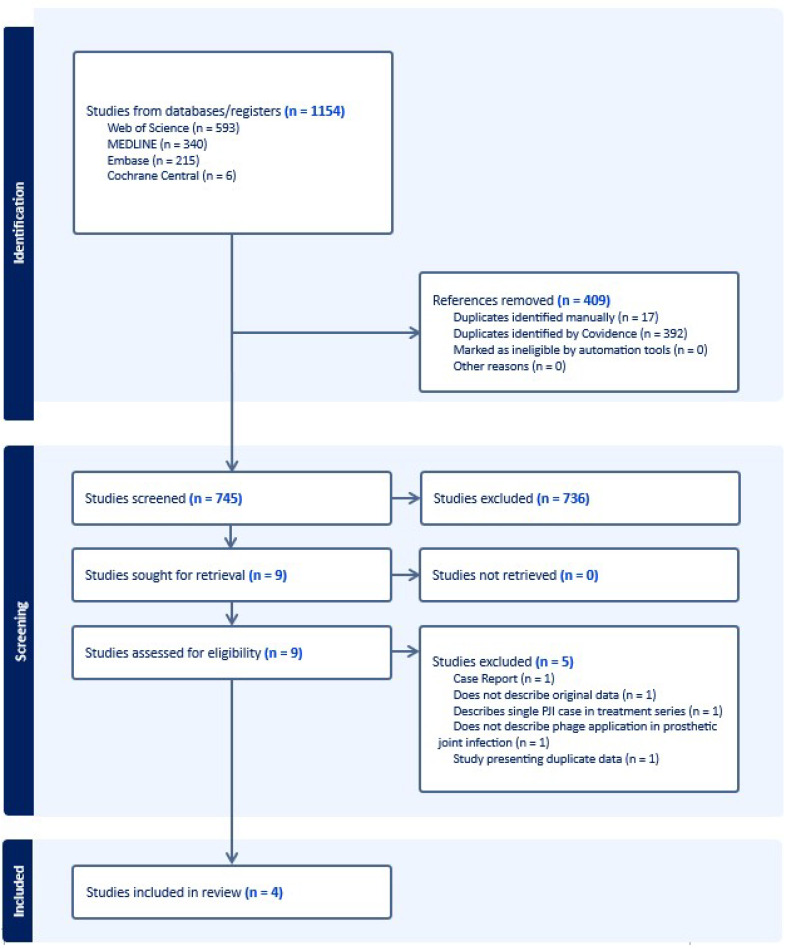
PRISMA diagram summarizing article search and screening strategy.

**Figure 2 medicina-60-00790-f002:**
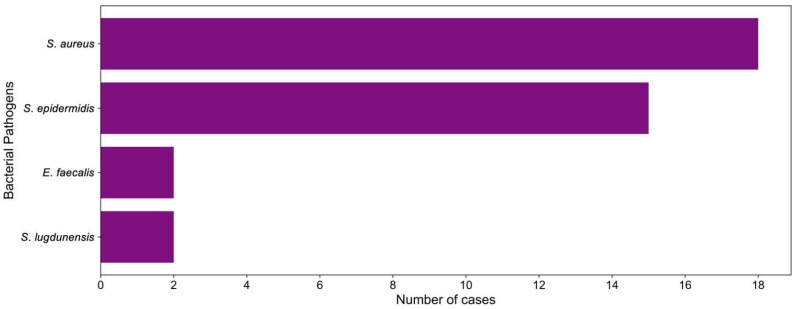
Bacterial pathogens implicated in the included cases treated with phage therapy.

**Figure 3 medicina-60-00790-f003:**
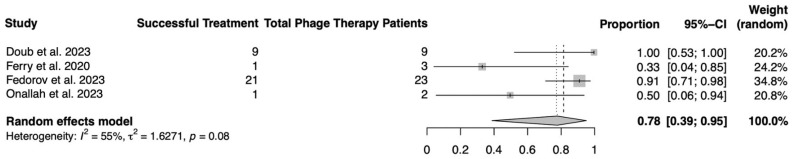
Random-effects proportional meta-analysis of phage treatment for prosthetic joint infections.

**Figure 4 medicina-60-00790-f004:**
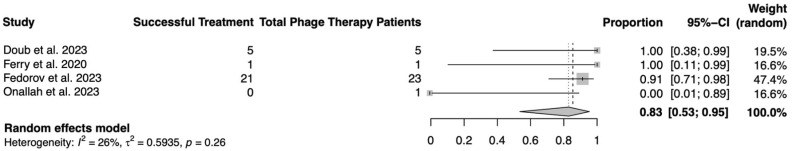
Subgroup random-effects proportional meta-analysis of phage treatment for prosthetic joint infections with minimum 12-month follow-up.

**Table 1 medicina-60-00790-t001:** Summary of included studies.

Study	Study Country	Study Design	Mean Age, Years	Number of Patients, N (Number Female)	Phage Therapy Group, N	Control Group, N	Previously Failed Conventional Therapy?	Included Indications (N)	Surgical Procedure Performed with Phage Therapy (DAIR, 1-Stage, 2-Stage) * (N)	Bacterial Pathogens Implicated in Patients Receiving Phage Therapy (N)	Bacteriophage Regimen (Number of Patients Receiving Regimen)	Bacteriophage Manufacturer	Route of Phage Administration (N)	Duration of Phage Therapy, Days	Administered with Antibiotics? (Y/N) (N)	Reported Outcome (N)	Follow-Up Period	Adverse Events from Phage Therapy (N)
Fedorov et al., 2023 [29]	Russia	Prospective Case-Control	56	45 (not specified)	23	22	Not specified	Hip PJI (45)	1-Stage (45)	MSSE (8) MRSE (6) MSSA (8) MRSA (1)	“Staphylococcal bacteriophage” cocktail (23)	Microgen	Intraarticular (23)	10	Yes (23)	Resolution of PJI signs and symptoms (21) Infection Persistence (2)	12 Months	Transient fever (2)
Doub et al., 2023 [25]	USA	Case Series	Not specified	9 (not specified)	9	N/A	All failed previous surgical and antibiotic therapy	Knee PJI (7) Hip and knee PJI (2)	DAIR (6) 1-Stage or 2-Stage (4)	MRSA (5) *S. lugdunensis* (2) *E. faecalis* (1) *S. epidermidis* (1)	EF 1 (1) PM448 (1) Mallokai (5) SawIQ0488ø1 (1) SaGR5Φ1 (1)	Yale Center for Phage Biology and Therapy Adaptive Phage Therapeutics	Intraarticular (9) Intravenous (8)	1–5	Yes (9)	Resolution of PJI signs and symptoms (9) **	5 months–2.5 years	Asymptomatic transaminitis (5)
Ferry et al., 2020 [26]	France	Case Series	82	3 (1)	3	N/A	All failed previous surgical and antibiotic therapy	Knee PJI (3)	DAIR (3)	MSSA (3)	One or more of the following: PP1493, PP1815, and PP1957	Pherecydes Pharma library	Intraarticular (3)	Single dose at end of DAIR procedure	Yes (3)	Resolution of PJI signs and symptoms (1) Infection Persistence (3) ***	7–30 months	None reported
Onallah et al., 2023 [19]	Israel	Case Series	65	2 (2)	2	N/A	Not specified	Hip PJI (1) Prosthetic joint infection, unspecified (1)	Not Specified (2)	*E. faecalis* (1) MSSA (1)	EFGrNG + EFGrKN (1) SaWIQ493Ph1 (1)	Queen Astrid Military Hospital Adaptive Phage Therapeutics	Intravenous (2)	Not specified, though between 6 and 45 days	Yes (1) No (1)	Resolution of PJI signs and symptoms (1) Infection Persistence (1)	3 months–1 year	None reported

* DAIR = Debridement, Antibiotics, and Implant Retention; 1-Stage = Single Stage Revision; 2-Stage = Two Stage Revision; MSSE = Methicillin-Sensitive *S. epidermidis*; MRSE = Methicillin-Resistant *S. epidermidis*; MSSA = Methicillin-Sensitive *S. aureus*; MRSA = Methicillin-Resistant *S. aureus*. ** Authors report one patient sustained a subsequent, new infection of the treated total joint from an infected port, ultimately requiring arthrodesis, while another required amputation as a complication of poor wound healing and soft tissue contractures, though cultures at time of amputation were negative. *** Authors report one patient had persistent drainage after phage therapy and underwent repeat DAIR with resolution of symptoms after the second procedure.

## Data Availability

The datasets used and/or analyzed during the current study are available from the corresponding author on reasonable request.

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
