# Peer review of "How Effective Is Phage Therapy for Prosthetic Joint Infections? A Preliminary Systematic Review and Proportional Meta-Analysis of Early Outcomes"

_medicina, 2024, doi:10.3390/medicina60050790_

Round 1

Reviewer 1 Report

Comments and Suggestions for Authors

1. Line 37. The cost is $50,000 in the USA, it is worth clarifying.
2. Y, SL and NM are referred to as reviewers in the manuscript. If JY, SL and NM are named reviewers, then who are the co-authors of the manuscript?
3. Figure 1. What are the criteria for excluding 736 manuscripts?
4. It is not clear why individual case reports were excluded, i.e. 2 cases in the manuscript is ok, and 1 case is not enough. Results: 4 manuscripts were analyzed, and 21 patients (65%) of 32 were patients from 1 manuscript. Is this enough to call the study "Systematic Review"? Formally yes, it is enough if the methodology was followed...

Reviewer 2 Report

Comments and Suggestions for Authors

Dear Authors,

Thank you for the opportunity to review the study "How Effective is Phage Therapy for Prosthetic Joint In3 infections? A Preliminary Systematic Review and Proportional Meta-Analysis of Early Outcomes".

The manuscript is clear, relevant to the field covered by the MEDICINE journal and the study is presented in a way that is understandable to the reader.

The included literature is appropriately matched to the manuscript, which increases the quality.

The manuscript has a clear purpose and presents both the methodology and results of the study in an understandable way.

The table and figures are presented in a clear and understandable way.

To sum up, the work is prepared in a way that is understandable to read and easy to interpret in a given field. It seems to be ready for publication, which may contribute to greater reader citation.

Author Response

We thank the reviewer for their kind words and effort and time evaluating our manuscript. 

Reviewer 3 Report

Comments and Suggestions for Authors

Antimicrobial resistance is the new pandemic, therefore, treatments that help eradicate infections are relevant. The authors conducted a systematic review on Phage Treatment for Prosthetic Joint Infections (PJI). For this, they realized a compilation of 4 published papers for provide a preliminary assessment of early phage therapy treatment outcomes for cases of PJI. The work considers the inclusion and exclusion criteria of publications, describes the search and data extraction strategies and the detection of methodological biases. The main limitations are a very small number (n=4) of works published on the subject and a low quality of evidence (Fair and Poor rating). The references are adequate and current, most of the citations correspond to the last 5 years. Other questions: The objective is worded differently in the abstract and introduction. It is suggested to unify throughout the manuscript. Materials and methods must be written in the past tense (see archive).
